# Rationale for Combining the BCL2 Inhibitor Venetoclax with the PI3K Inhibitor Bimiralisib in the Treatment of IDH2- and FLT3-Mutated Acute Myeloid Leukemia

**DOI:** 10.3390/ijms232012587

**Published:** 2022-10-20

**Authors:** Katja Seipel, Yvo Brügger, Harpreet Mandhair, Ulrike Bacher, Thomas Pabst

**Affiliations:** 1Department for Biomedical Research (DBMR), University of Bern, 2008 Bern, Switzerland; 2Department of Hematology, University Hospital Bern, 3010 Bern, Switzerland; 3Department of Medical Oncology, University Hospital Bern, 3010 Bern, Switzerland

**Keywords:** acute myeloid leukemia (AML), B-cell lymphoma-2 (BCL2), venetoclax (ABT-199), isocitrate dehydrogenase 2 (IDH2), bimiralisib (PQR309), mammalian target of rapamycin (mTOR), phosphoinositide 3-kinase (PI3K), Fms-related receptor tyrosine kinase 3 (FLT3), protein tyrosine phosphatase non-receptor type 11 (PTPN11)

## Abstract

In October 2020, the FDA granted regular approval to venetoclax (ABT-199) in combination with hypomethylating agents for newly-diagnosed acute myeloid leukemia (AML) in adults 75 years or older, or in patients with comorbidities precluding intensive chemotherapy. The treatment response to venetoclax combination treatment, however, may be short-lived, and leukemia relapse is the major cause of treatment failure. Multiple studies have confirmed the upregulation of the anti-apoptotic proteins of the B-cell lymphoma 2 (BCL2) family and the activation of intracellular signaling pathways associated with resistance to venetoclax. To improve treatment outcome, compounds targeting anti-apoptotic proteins and signaling pathways have been evaluated in combination with venetoclax. In this study, the BCL-XL inhibitor A1331852, MCL1-inhibitor S63845, dual PI3K-mTOR inhibitor bimiralisib (PQR309), BMI-1 inhibitor unesbulin (PTC596), MEK-inhibitor trametinib (GSK1120212), and STAT3 inhibitor C-188-9 were assessed as single agents and in combination with venetoclax, for their ability to induce apoptosis and cell death in leukemic cells grown in the absence or presence of bone marrow stroma. Enhanced cytotoxic effects were present in all combination treatments with venetoclax in AML cell lines and AML patient samples. Elevated in vitro efficacies were observed for the combination treatment of venetoclax with A1331852, S63845 and bimiralisib, with differing response markers for each combination. For the venetoclax and bimiralisib combination treatment, responders were enriched for *IDH2* and *FLT3* mutations, whereas non-responders were associated with *PTPN11* mutations. The combination of PI3K/mTOR dual pathway inhibition with bimiralisib and BCL2 inhibition with venetoclax has emerged as a candidate treatment in *IDH2-* and *FLT3*-mutated AML.

## 1. Introduction

With a median age of 68 years at diagnosis, acute myeloid leukemia (AML) is predominantly a disease of the elderly. The majority of these patients are not eligible for intensive chemotherapy with curative intent, and the standard of care has been treatment with hypomethylating agents (HMA) [1,2]. The addition of the BCL2 inhibitor venetoclax (ABT-199) to the HMA backbone leads to an increase in response and survival rates [3]. In October 2020, the Food and Drug Administration granted regular approval to venetoclax (VENCLEXTA^®^, AbbVie Inc., North Chicago, IL, USA and Genentech Inc., San Francisco, CA, USA) in combination with HMA including azacitidine, decitabine, or with low-dose cytarabine (LDAC) for newly-diagnosed AML in adults 75 years or older, or in patients with comorbidities precluding intensive chemotherapy. The combination of venetoclax and the HMA azacitidine results in a remission rate of approximately 70% [3]. However, a significant number of patients have refractory disease, and the majority of responding patients will ultimately relapse.

High response rates and durable remissions to venetoclax treatment were found to be associated with *NPM1*, *DNMT3A*, or *IDH2* mutations, while resistance was associated with mutations in *FLT3*, *RAS*, or *TP53* genes [4,5,6]. In another study, responders were enriched for *TET2*, *IDH1*, and *IDH2* mutations, while non-responders were associated with *FLT3* and *RAS* mutations [7]. *RUNX1* and *SRSF2* mutations may also be associated with venetoclax response [8,9]. Different mechanisms such as deletion or inactivation of TP53 but also increased expression of MCL-1, another active pro-survival member of the BCL-2 family, were reported to be related to venetoclax resistance [10].

Hematological cells of various origins, including AML, exhibit specific dependencies on either BCL-2, BCL-XL, or MCL-1 for survival [11,12]. This dependency may be associated with selective sequestration of the pro-apoptotic proteins BIM, BAX, and BAK by the specific anti-apoptotic BCL-2 protein. BH3-mimetics displace pro-apoptotic BH3-containing proteins from their anti-apoptotic target. The BCL-2 inhibitor venetoclax induced BAX-dependent apoptosis, while the MCl-1 inhibitor S63845 induced mainly BAK-dependent apoptosis. S63845 displayed impressive potency at low nanomolar concentrations in preclinical in vitro and in vivo models of hematological malignancies, including MM, AML, CML, and c-MYC-driven Burkitt lymphoma [13]. S63845 has been proposed as a candidate treatment in AML in combination with the MEK inhibitor trametinib or the BMI1 inhibitor PTC596 in preclinical studies [14,15]. Clinical studies of combination treatments with MCL-1 inhibitors and venetoclax have been initiated in hematological malignancies (NCT03672695, NCT04702425). Venetoclax in combination with trametinib has been proposed as targeted therapy in *RAS*-mutated AML [16], combinations of STAT5- and MCL-1 inhibitors in *FLT3-* or *TET2*-mutated AML [17]. In vivo administration of cobimetinib in combination with venetoclax demonstrated anti-leukemia efficacy in acute myeloid leukemia xenograft mouse models [18], and a phase 1/2 study combining the MEK inhibitor cobimetinib with venetoclax in AML has been initiated (NCT02670044). In lymphatic leukemias combinations of venetoclax and BCL-XL inhibitor A1331825 were proposed for treatment in B-lineage acute lymphatic leukemia (B-ALL) [19], while venetoclax and PI3K inhibitors are employed in the treatment of chronic lymphatic leukemia (CLL) [20,21].

In myeloid cells, FLT3 is a growth factor receptor signaling via PI3K-AKT-mTOR, and *FLT3* gene mutations lead to abnormal activation of the pathway in AML [22,23]. About 50–80% of AML patients display constitutive PI3K/Akt/mTOR activation, and this was associated with reduced survival [24]. Pan PI3K inhibitors are expected to reduce the risk of drug resistance that might occur in case of treatment with compounds targeting a single PI3K isoform and, together with mTOR inhibition, could prevent feedback loop of AKT activation following mTOR inhibition. Co-targeting BCL-2 and PI3K may induce apoptosis in AML cells [25]. PQR309 (bimiralisib) is a novel orally bioavailable selective dual PI3K/mTOR inhibitor [26]. Bimiralisib alone or in combination with venetoclax has been evaluated in preclinical lymphoma setting [27] and in a phase II study in R/R lymphoma [28]. Unesbulin (PTC596) is a second-generation BMI-1 inhibitor that downregulates MCL-1 expression in AML cells and may influence expression of MCL1 inducers including MEK, ERK, AKT, STAT3, and STAT5 [29]. The combination of PTC596 and S63845 or trametinib may be an effective treatment in CD34+ adverse risk AML [15]. Chemotherapies may be effective at eradicating leukemic cells in the peripheral blood, but not in the bone marrow niche, where leukemic cells are sheltered [30]. Valuable information may be gained in the preclinical evaluation of novel therapies when AML cells are grown in the presence of bone marrow stroma [17].

In the current study, the BCL-XL inhibitor A1331852, the MCL1-inhibitor S63845, the PI3K inhibitor bimiralisib (PQR309), the BMI-1 inhibitor unesbulin (PTC596), the MEK-inhibitor trametinib (GSK1120212), and the STAT3 inhibitor C-188-9 were assessed as single agents and in combination with venetoclax, for their ability to induce apoptosis and cell death in leukemic cells grown in the absence or presence of bone marrow stroma. Several combinations were found to be effective in vitro: venetoclax and S63845, venetoclax and A1331825, and venetoclax and PQR308.

## 2. Results

### 2.1. Variable Susceptibility of AML Cell Lines to Venetoclax and Various Targeted Therapies

To determine the sensitivity of AML cells to different targeted compounds, AML cells were subjected to in vitro cytotoxicity assays. Seven AML cell lines and one BCL2-driven DLBCL cell line were treated for 20 h in dose escalation experiments before cell viability assessment. Our panel of AML cell lines covered the majority of the morphologic and molecular subtypes, particularly including *FLT3*-ITD and *FLT3* wild type, *NPM1* mutant and wild type, as well as *TP53* wild type, mutant, hemizygous, and null cells (Table 1).

We observed that the BCL2-driven and *TP53* double mutant lymphoma cell line OCI-Ly1 was highly susceptible to venetoclax with IC50 of 60 nM. The AML cell lines ML-2 and MOLM-13 were susceptible to venetoclax with IC50 of 100 and 200 nM, while OCI-AML3, SKM-1, and HL-60 had IC50 of 600 nM, 1 μM, and 1.6 μM, respectively. PL-21 and MOLM-16 cells were resistant to venetoclax with IC50 > 10 μM (Appendix A, Appendix A, Table 2). The IC 50 levels of susceptible AML cells were in the range of physiological relevant concentrations. Venetoclax steady state plasma concentrations of 1.2–3.5 μM were observed in CLL patients receiving the recommended phase 2 dose of 400 mg per day [31].

In order to find effective treatment combinations, we focused on inhibitors with the potential to elicit synergistic effects in combination with venetoclax. These included targeted compounds tested in previous studies like BMI1-, MCL1-, and MEK- inhibitors [8,9] as well as BCL-XL-, PI3K- and STAT3 inhibitors as indicated in Figure 1. With respect to the MCL-1 inhibitor S63845, we found that PL-21 and MOLM-16 cells were resistant with IC50 > 10 μM, while the other AML cell lines were susceptible with IC50 of 100–500 nM. With respect to the BCL-XL inhibitor A1331825, PL-21 and MOLM-16 cells were susceptible with IC50 of 2–4 μM, while other AML cell lines were resilient with IC50 values of 5–10 μM. With respect to the dual PI3K-mTOR inhibitor bimiralisib, the *PTEN* deleted OCI-Ly1 cells were most sensitive with an IC50 of 1 μM, the *FLT3*-mutated MOLM-13 with IC50 of 2 μM, ML-2 and SKM-1 with IC50 of 3 μM, while OCI-AML3, PL-21 and MOLM-16 were resilient with IC50 of 10 μM. With respect to the BMI-1 inhibitor PTC596, ML-2, PL-21 and MOLM-16 cells had IC50 >1 μM, while the other AML cell lines were susceptible with IC50 in the range of 200–500 nM. With respect to the MEK inhibitor trametinib, HL-60, ML-2, MOLM-13, OCI-AML3 and SKM-1 cells were most susceptible with IC50 of 80–120 nM, while MOLM-16 and PL.-21 cells were resistant with IC50 of 10 μM. The IC50 levels of susceptible AML cells were in the range of physiologically relevant concentrations: Bimiralisib plasma levels were 2 μM in patients treated for solid tumors [32]; A-1331852 plasma levels were 2 μM in rat p.o. [33]; PTC-596 cmax ranged from 1 to 5 μM in patients with advanced tumors [34]; trametinib plasma levels were 20 nM in patients treated for BRAF melanoma [35]. Only for the STAT3 inhibitor C-188-9 the in vitro IC50 levels of AML cells exceeded the 2 μM plasma concentration determined in PDX mice [36]. The in vitro IC50 of C-188-9 was in the range of 4–8 μM in AML cell lines, and in the range of 8–18 μM in primary AML samples [37].

### 2.2. Synergistic Effects on Cell Viability in AML Cell Lines Treated with Venetoclax Combinations

Cell viability was determined in seven AML cell lines and one BCL2-driven DLBCL cell line treated with increasing dosages of single compounds and in combination using a variety of targeted therapies including the BCL-XL inhibitor A1331825, the PI3K inhibitor bimiralisib (PCR309), the STAT3 inhibitor C-188-9, the BMI-1 inhibitor PTC596, the MCL1 inhibitor S63845, and the MEK inhibitor trametinib. Drug concentrations in the combination studies were chosen to correspond to minimally effective concentrations in single compound assays determined in initial titration. Five AML cell lines and OCI-Ly1 were susceptible to 100 nM venetoclax and multiple combination treatments, while PL-21 and MOLM-16 cells were resistant to 100 nM venetoclax and most combination treatments, with the exception of venetoclax and A1331825 (Figure 2). Across the panel of AML cell lines, cytotoxic effects were enhanced in the combination treatments, for venetoclax and bimiralisib (*p* = 0.002), trametinib (*p* = 0.003), S63845 (*p* = 0.005), A1331825 (*p* = 0.01), PTC596 (*p* = 0.01), or C-188-9 (*p* = 0.04), Combination indexes were calculated according to Chou Talalay [38]. Synergistic effects on cell viability were calculated to be moderate to strong for venetoclax combined with S63845, A1331825, trametinib or bimiralisib, and mild to moderate for venetoclax combined with PTC596 or C-188-9 (Table 3).

### 2.3. Altered Susceptibility to Targeted Therapies in AML Cells Grown in the Presence of Bone Marrow Stroma

To investigate the efficacy of venetoclax combination treatments in the bone marrow environment, cell viability was determined in the susceptible AML cell lines MOLM-13, ML-2, SKM-1, and OCI-AML3, grown in the absence or presence of bone marrow stroma cells. Stroma cells secrete granulocyte and macrophage colony-stimulating factors (G-CSF, GM-CSF, M-CSF) and a variety of cytokines, which can induce STAT signaling in leukemic cells [39,40,41,42]. AML cells grown in the presence of bone marrow stroma were generally less affected by the combination treatment than AML cells grown in the absence of stroma, indicating a protective effect of the bone marrow environment on AML cells (Figure 3). We observed that only the combination of venetoclax and MCL1 inhibitor S63845 induced cell death with equal efficacy in AML cells grown in the absence or presence of bone marrow stroma. MOLM-13 cells were protected toward venetoclax and combination treatments when grown on stroma (Figure 3A), ML-2 cells appeared to be protected toward trametinib when grown on stroma (Figure 3B). OCI-AML3 cells were more susceptible to venetoclax when grown on stroma, but protected toward all venetoclax combination treatments (Figure 3C). SKM-1 cells were more susceptible to A1331825, and protected toward bimiralisib and trametinib, when grown on stroma (Figure 3D).

### 2.4. Venetoclax Combination Treatment Induces Cell Cycle Arrest, Apoptosis and Cell Death

Effects of venetoclax combination treatments on cell cycle and apoptosis were studied in the susceptible AML cell lines MOLM-13, ML-2, SKM-1, and OCI-AML3. Treatment with bimiralisib and venetoclax lead to the induction of apoptosis, cell cycle arrest, and cell death in MOLM-13 cells (Figure 4, Appendix A), as well as in SKM-1 (Appendix A) and ML-2 cells (Appendix A). Venetoclax treatment lead to an increase in dead cells (Figure 4B), reduction of vital cells (Figure 4E) and increase in apoptotic cells (Figure 4F and Appendix A). Bimiralisib treatment leads to the induction of G1 cell cycle arrest (Figure 4C and Appendix A). The induction of apoptosis and cell death, as well as reduction of vital cells was significantly enhanced in MOLM-13 cells treated with venetoclax and bimiralisib (Figure 4B,E). The induction of apoptosis and cell death, as well as reduction of vital cells was significantly enhanced in ML-2 and OCI-AML3 cells treated with venetoclax and A1331825 or C-188-9 (Appendix A).

### 2.5. Venetoclax Combination Treatments with Differential Efficacy in Subsets of AML Patients

After initial studies in AML cell lines, the treatment combinations of venetoclax with A1331825, S63845, bimiralisib, trametinib, C-188-9, or PTC596 were applied to patient-derived mononuclear cells isolated from peripheral blood (PBMC) or bone marrow (BMMC) (Table 4). A total of 26 AML, one CML, two NHL, as well as PBMCs of four healthy donors (HD) were subjected to single compound and combination treatments. The tested combination treatments induced minor reduction of cell viabilities in mononuclear cells isolated from healthy donors, and substantial reduction of cell viability in 50–70% of AML samples treated with venetoclax monotherapy (Figure 5A) or in combination with A1331825, S63845 or bimiralisib (Figure 5B–D). Venetoclax in combination with trametinib, C-188-9 or PTC-596 was less effective with substantial reduction of cell viability in 20–40% of AML samples (Figure 5E,F). The patient samples were divided in two subgroups of similar size, one with major (strong) response (SR), and one with minor (normal) response (NR). The median cell viabilities in the SR groups were 70% in 100 nM venetoclax treatment (Figure 5A), 50% in the venetoclax combination with 1 μM A1331825 or 1 μM bimiralisib (Figure 5B,C), 43% in combination with S63845 (Figure 5D), 55% in combination with 100 nM trametinib (Figure 5E), 60% in combination with 1 μM C-188-9 (Figure 5F), and 62% in combination with 200 nM PTC596 (Figure 5G).

Potential response markers were deduced from the correlation analysis of cell viabilities grouped according to diagnostic parameters including gene mutation status, peripheral blood and bone marrow blast cells percentage, and CD34 positivity. In venetoclax-treated AML, the presence of *IDH2* mutation, as well as elevated blast cell percentage in peripheral blood or bone marrow associated with response, presence of *PTPN11, TET2* or *ASXL1* mutation indicated lack of response, while *FLT3*, *NPM1, RUNX1,* and *TP53* status as well as CD34 levels were not associated with treatment response (Figure 6). In the venetoclax and A1331825 combination treated AML gene mutation status of *TET2, FLT3*, and *TP53* as well as CD34, levels were inconsequential, *IDH2* mutation and elevated blast cell percentage were associated with response, while the presence of *PTPN11 or ASXL1* mutation indicated lack of response (Figure 7). In the venetoclax and bimiralisib combination treatment, the presence of *IDH2 or FLT3* mutation and elevated blast cell counts were indicators of response, *ASXL1* and *TP53* status was inconsequential, while presence of *PTPN11* or *TET2* mutation indicated a lack of response (Figure 8). In the combination treatment of venetoclax and S63845, the presence of *IDH2* mutation and elevated blast counts were associated with response, presence of *TET2* mutation indicated lack of response, while *NPM1, FLT3*, *ASXL1, PTPN11*, and *TP53* status were inconsequential (Figure 9). Notably, AML patient samples with *TET2* mutations carried additional mutations in *PTPN11, KRAS, FLT3,* or *TP53* genes, all of which may be associated with venetoclax resistance (Table 4). In the combination treatments of venetoclax and trametinib, C-188-9 or PTC596, the number of samples with reduced cell viability was small, and response markers were not identified (Appendix A).

## 3. Discussion

Treatment response to the BCL2 inhibitor venetoclax together with hypomethylating agents may be short-lived with leukemia relapse as the major cause of treatment failure. Multiple studies have indicated that the upregulation of other anti-apoptotic proteins of the B-cell lymphoma 2 (BCL2) family and the activation of intracellular signaling pathways were the major factors leading to resistance to venetoclax [4,5,7]. Accordingly, targeting anti-apoptotic proteins BCL-XL and MCL-1 as well as targeting signaling pathways leading to the induction of BCL-XL and MCL-1 may enhance and prolong treatment response to the BCL-2 inhibitor venetoclax.

In this study, we describe a panel of venetoclax combination treatments with enhanced cytotoxic effects on AML cells grown in the absence or presence of bone marrow stroma, including the BCL-XL inhibitor A133825, the MCL1 inhibitor S63845, the BMI1 inhibitor PTC596, the dual PI3K-mTOR inhibitor bimiralisib, the STAT3 inhibitor C-188-9, and the MEK inhibitor trametinib. The in vitro IC50 concentrations of the tested inhibitors were determined to be in the range of physiologically relevant concentrations for all compounds, except C-188-9, where in vitro IC50 levels exceeded the plasma concentrations present in PDX mice. AML cells grown in the presence of bone marrow stroma were generally less affected by the combination treatments than AML cells grown in the absence of stroma, indicating a protective effect of the bone marrow environment on AML cells. MOLM-13 cells were protected toward venetoclax and combination treatments when grown on stroma, ML-2 cells were protected toward trametinib, OCI-AML3 cells were protected toward all venetoclax combination treatments, and SKM-1 cells were more susceptible to venetoclax and A1331825, and protected toward bimiralisib and trametinib, when grown on stroma. SKM-1 cells have been found to be more susceptible to venetoclax and to the STAT5 inhibitor AC-4-130 when grown on stroma [15]. A cell-type-specific dependence on STAT5 signaling may cause elevated susceptibility to the BCL-2- and BCL-XL inhibitors and a reduced response to PI3K- and MEK inhibitors, in the bone marrow environment. Various cellular components, cytokines, and chemokines present in the bone marrow may impact AML initiation and therapy resistance at the cellular and molecular level [43,44]. We found that only the combination of venetoclax and MCL1 inhibitor S63845 induced cell death with equal efficacy in AML cells grown in the absence or presence of bone marrow stroma, indicating a potential advantage of applying this combination in the treatment of AML, as this may eradicate leukemic stem cells in the bone marrow. A synergistic effect of S63845 toward venetoclax-mediated apoptosis of AML cells in the context of interaction with the BM microenvironment that intrinsically mediates resistance to BCL2 inhibition has been previously described [45]. Targeting MCL-1 may dysregulate the cellular metabolism and leukemia–stroma interactions and re-sensitize acute myeloid leukemia to BCL-2 inhibition [46].

To validate the findings in a translational setting, venetoclax combination treatments were applied to mononuclear cells isolated from the peripheral blood or bone marrow of primary AML patients. The addition of the BCL-XL inhibitor A133825, the MCL1 inhibitor S63845 or the PI3K inhibitor bimiralisib to venetoclax induced substantial reductions of cell viability in 50–70% of the tested AML samples, while the addition of the MEK inhibitor trametinib, the BMI-1 inhibitor PTC596 or the STAT3 inhibitor C-188-9 to venetoclax was less effective, with substantial reductions of cell viability in 20–40% of tested AML samples. In order to define the patient subgroups who may profit from novel targeted combination therapies, potential response markers were deduced from the correlation analysis of cell viabilities grouped according to diagnostic parameters, including gene mutation status of prevalent tumor suppressors and oncogenes, peripheral blood and bone marrow blast cell percentage, and levels of CD34 positive cells. A significant association between venetoclax response and elevated blast cell percentage has been previously described [47], and was reproduced in our study with a boundary value of 60% peripheral blood blast percentage, and 70% bone marrow blast cell infiltration. Mononuclear cells isolated from AML patients with elevated blast cell percentage were more susceptible to venetoclax and multiple venetoclax combinations in vitro. *IDH2, NPM1, FLT3, DNMT3A, PTPN11, ASXL1, TET2, KRAS, RUNX1*, and *TP53* genes have been described as response markers to venetoclax treatment [4,5,7,8]. In our study, *IDH2* mutation was the single most significant and consistent biomarker associated with response to venetoclax and multiple venetoclax combination treatments, while *TP53* mutation was not associated with response. Notably, the *TP53* double mutant cell line OCI-Ly1 was most sensitive to venetoclax, while the *TP53* double mutant cell line MOLM-16 was resistant.

The mutation status of other genes was relevant to response, however, with differential indicators in specific venetoclax combination treatments. *FLT3* mutations were associated with response to venetoclax and bimiralisib, but not to venetoclax monotherapy or in combination with A1331825 or S63845, indicating that *FLT3*-mutated cells may be specifically susceptible to the combined inhibition of BCL2, PI3K, and mTOR. The *TET2* gene mutation has been associated with response to venetoclax and HMA combination treatment [5]. In our study, *TET2* mutation apparently associated with resistance to venetoclax in combination with A1331825 or bimiralisib, possibly due to presence of concurrent mutations in *KRAS, PTPN11*, or *TP53* genes in the primary AML samples. *PTPN11* mutations were associated with lack of response to venetoclax and A1331825 or bimiralisib, but not to venetoclax and S63845, indicating differential target cell specificity and differential mechanisms of action for S63845 in combination with venetoclax. Activating mutations of the SHP2 protein, encoded by the *PTPN11* gene, leads to hyper-activation of the downstream RAS-MAPK signaling pathway and confer resistance to venetoclax and multiple venetoclax combinations. The combination of venetoclax and the MCL1 inhibitor AZD5991 was proposed to overcome this resistance [47]. In accordance, the combination treatment with venetoclax and the MCL-1 inhibitor S63845 resulted in reduced cell viabilities in primary AML samples in vitro independent of *PTPN11* status. *ASXL1* mutations have been associated with distinct epigenomic alterations that lead to sensitivity to venetoclax and azacytidine [48]. In our study, *ASXL1* mutation apparently associated with resistance to venetoclax in combination with A1331825, but not in combination with S63845 or bimiralisib, possibly due to the presence of concurrent mutations in *KRAS, PTPN11, RUNX1,* or *TP53* genes in the primary AML samples. Further studies in larger cohorts may be required to validate the relevance of *TET2*, *TP53*, and *ASXL1* mutation status as response markers to venetoclax combination treatments. Additional biomarkers of response to venetoclax combination treatments may arise from studies on expression levels of pro-apoptotic BCL-2 family proteins and anti-apoptotic proteins (BIM, BAX, and BAK) as well as the components of the upstream signaling pathways in larger cohorts.

In conclusion, different combinations of targeted therapies emerge that are suitable in the treatment of specific subsets of AML patients. In this study, elevated in vitro efficacies were detected in the combination treatments of venetoclax with BH3 mimetics A1331852 and S63845 and the dual PI3K-mTOR inhibitor bimiralisib, in AML with elevated blast cell percentage, at drug concentrations that can be reached in vivo in the plasma, with different response markers in each combination. The combination treatment of venetoclax and A1331825 may be effective in the treatment of *IDH2*-mutated AML in the absence of *PTPN11* mutations, while the combination of venetoclax and S63845 may be effective in AML in the presence of *PTPN11* mutations. The combination treatment of venetoclax and bimiralisib may be effective in AML with *IDH2, NPM1*, and *FLT3* mutations. We propose the combination of PI3K/mTOR dual pathway inhibition with bimiralisib and BCL2 inhibition with venetoclax as a candidate treatment in clinical trials for *IDH2-* or *FLT3*-mutated AML.

## 4. Materials and Methods

### 4.1. Patient Samples

Mononuclear cells of AML patients diagnosed and treated at the University Hospital, Bern, Switzerland, between 2018 and 2022, were included in this study. Informed consent from all patients was obtained according to the Declaration of Helsinki, and the studies were approved by decisions of the local ethics committee of Bern, Switzerland, decision number 221/15. Peripheral blood mononuclear cells (PBMCs) and bone marrow mononuclear cells (BMMCs) were collected at the time of diagnosis before initiation of treatment. The AML cells were analyzed at the central hematology laboratory of the University Hospital Bern according to state of the art techniques [49]. Mutational screening for *FLT3, NPM1, TP53*, and conventional karyotype analysis of at least 20 metaphases were performed in all samples. In addition, all samples were analyzed by NGS sequencing of the myeloid panel genes. The genes tested within the NGS panel can be categorized into several major functional categories, including the spliceosome (U2AF1, SF3B1, SRSF2, and ZRSR2), epigenetic modifiers (TET2, DNMT3A, BCOR, ASXL1, IDH1, and IDH2), cohesions (STAG2, RAD21, and SMC3), transcription factors (TP53, RUNX1, WT1, and ETV6), signaling molecules (NF1, NRAS, CBL, PTPN11, JAK2, and FLT3), and chromatin modifiers (EZH2 and ASXL1).

### 4.2. Cell Lines and Cell Culture

Human AML cells lines OCI-AML3 (AML-M4, FLT3wt, DNMT3A R882C, NPM1mut, and TP53wt), MOLM-13 (AML-M5, t(9;11), FLT3-ITD, and TP53wt), MOLM-16 (AML-M0, FLT3wt, TP53mut), ML-2 (AML-M4, t(6;11), FLT3wt, TP53mut) and HL-60 (AML-M2, FLT3wt, and TP53 null) as well as B-cell lymphoma cell line OCI-Ly1 (B-NHL, t(14;18), PTEN del, TP53 mut) were supplied by the German Collection of Micro-organisms and Cell Cultures (DSMZ, Braunschweig, Germany). AML cells were grown in RPMI-1640 medium (R8758, SIGMA-ALDRICH, St. Louis, MO, USA), OCI-Ly1 in Iscove’s modified Dulbecco medium (I3390, SIGMA-ALDRICH, St. Louis, MO, USA), supplemented with 20% fetal bovine serum (F7524, SIGMA-ALDRICH, St. Louis, MO, USA), in tissue-culture flasks (REF 83.3911.502, Sarstedt, Nümbrecht, Germany) in a standard cell culture incubator at 37 °C with 5% CO_2_. Human bone marrow stroma cell line HS-5 (ATCC^®^ CRL-11882™) was supplied by the American tissue culture collection (ATCC, Manassas, VA, USA). HS-5 cells were grown in Dulbecco’s modified Eagle’s medium (D6064, SIGMA-ALDRICH, St. Louis, MO, USA) supplemented with 10% fetal bovine serum (F7524, SIGMA-ALDRICH, St. Louis, MO, USA) in standard tissue culture flasks (REF 83.3911.002, Sarstedt, Nümbrecht, Germany). HS-5 cells secrete granulocyte colony-stimulating factor (G-CSF), granulocyte-macrophage-CSF (GM-CSF), macrophage-CSF (M-CSF), Kit ligand (KL), macrophage-inhibitory protein-1 alpha, interleukin-1 alpha (IL-1alpha), IL-1beta, IL-1RA, IL-6, IL-8, IL-11, and leukemia inhibitory factor (LIF) [41,42]. For the co-culture assays HS-5 cells were plated on standard tissue culture plates (REF 83.3920, Sarstedt, Nümbrecht, Germany) on day 1. On day 2, Nunc 0.4 μm cell culture inserts (Thermo Fisher Scientific, Nunc A/S, Roskilde, Denmark) were placed over the HS-5 feeder layer and AML cells were filled into the cell culture inserts. On day 3, AML cells were collected from the six well inserts and replated on tissue culture plates suspension 96 well (REF 83.3924500, Sarstedt, Nünmbrecht, Germany), before addition of compounds. Cytotoxicity assays were performed on day 4.

### 4.3. Cytotoxicity Assays

For assays with AML cell lines, cells were plated at a density of 5 × 10^5^/mL on tissue culture plates suspension 96 well (REF 83.3924500, Sarstedt, Nümbrecht, Germany), and treated with targeted compounds. For assays with patient-derived mononuclear cells, the cells were cultured for 24 h prior to treatment. The BMI1 inhibitor PTC596 (HY-112041), the MCL1 inhibitor S63845 (HY-100741), the MEK inhibitor trametinib (HY-10999), the STAT3 inhibitor C-188-9 (HY-112288) and the PI3K inhibitor PQR309 (HY-12868) were purchased at MedChemExpress (Monmouth Junction, NJ, USA). PQR309 (bimiralisib) is a novel brain-penetrant dual PI3K/mTOR inhibitor with in vitro and in vivo anti-lymphoma activity as single agent and in combination. A stock solution of Venetoclax was prepared by dissolving a tablet in DMSO (Venclexta^®^, Abbvie Inc., North Chicago, IL, USA). Cell viability was determined 20 h after the start of treatment using the MTT-based cell proliferation kit I (Ref 11465007001, Roche Diagnostics GmbH, Mannheim, Germany). This time point was selected because the cellular responses were effectual for the calculation of combination indexes after 20 h of treatment with two compounds in leukemic cells. For AML cell lines, four independent assays (biological replicates) with four measurements (technical replicates) per dosage were performed. For hematological patient samples, two independent assays with three technical replicates per dosage were performed. For the calculation of combination indexes, two dosages of venetoclax and two dosages of the other compounds were applied alone and in combination. Combination indexes were calculated on Compusyn software (version 1.0; ComboSyn, Inc., Paramus, NJ, USA). Data are depicted as scatter plots with median values and SD. In grouped analysis, significance of differences in median values was calculated by Mann–Whitney test.

### 4.4. Imaging Cytometry

Imaging cytometry was carried out on the NC-3000 cell analyzer (ChemoMetec, Allerod, Denmark) with reagents supplied by ChemoMetec. To determine the induction of cell death apoptotic cells were stained with AnnexinV-CF488A conjugate (Biotium, Fremont, CA, USA) in AnnexinV buffer and Hoechst 33,342 (10 μg/mL) for 15 min at 37 °C, followed by several washes. Propidium iodide was added shortly before imaging. For cell cycle analysis, cells were incubated in lysis buffer with DAPI (10 μg/mL) for 5 min at 37 °C before imaging on the NC-3000 cell analyzer.

## Figures and Tables

**Figure 1 ijms-23-12587-f001:**
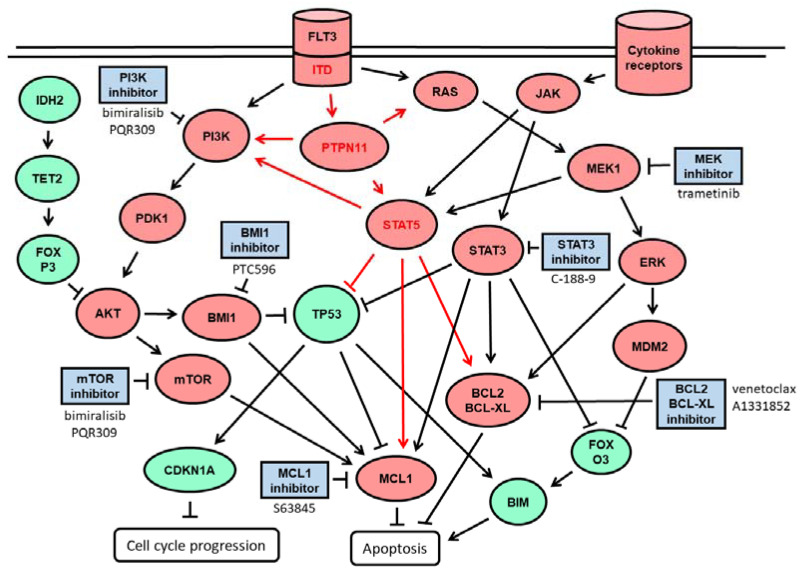
Schematic representation of signaling pathways in myeloid cells. The inducible growth factor receptor FLT3 signals via PI3K-AKT-mTOR and RAS-MEK-ERK (black arrows). FLT3-ITD, a constitutively active growth factor receptor, additionally induces PTPN11-STAT5 (red arrows). Activated cytokine receptors signal via Janus kinase (JAK)-signal transducer and activator of transcription (STAT) pathway. Signal transduction leads to inhibition of the tumor suppressor p53 and induction of the anti-apoptotic BH3 proteins BCL2, BCL-XL, and MCL1, thereby promoting proliferation and cell growth of myeloid cells. Oncogenic functions are indicated in red, tumor suppressor functions in green, chemical inhibitors in blue.

**Figure 2 ijms-23-12587-f002:**
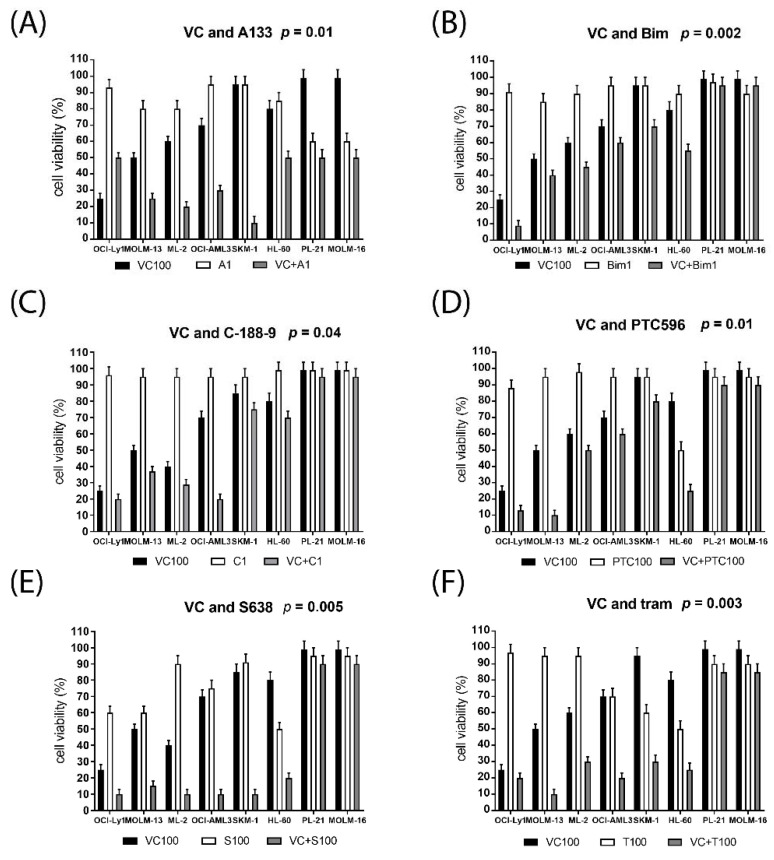
Susceptibility of leukemic cells to venetoclax combination treatment. Cell viability was determined in one lymphoma and seven AML cell lines treated for 20 h with single compounds and in combination with 100 nM venetoclax (VC) and 1 μM A1331825 (**A**), 1 μM bimiralisib (**B**), 1 μM C188-9 (**C**), 200 nM PTC596 (**D**), 100 nM S63845 (**E**), or 100 nM trametinib (**F**). Significance was calculated in a graph pad prism using grouped analysis with paired *t*-test comparing cell viabilities of VC treated and combination treated cells. A significance level of 0.05 indicates a 5% risk of concluding that a difference exists when there is no actual difference.

**Figure 3 ijms-23-12587-f003:**
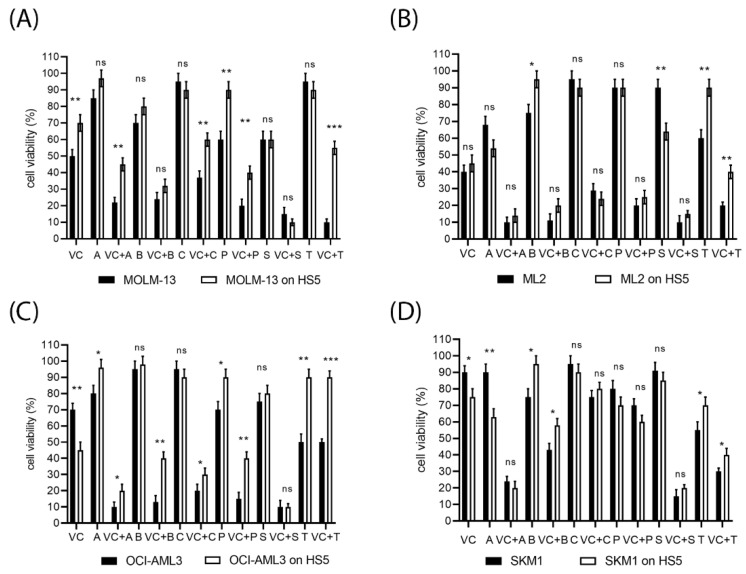
Susceptibility of AML cell lines to targeted therapies in the presence of bone marrow stroma. MOLM-13 (**A**), ML-2 (**B**), OCI-AML3 (**C**), and SKM-1 cells (**D**) were treated for 20 h with single compounds and in combination with venetoclax (VC), A1331825 (A), bimiralisib (B), C-188-9 (C), PTC596 (P), S63845 (S), or trametinib (T). Cell viability was determined in AML cells grown in the absence or presence of HS-5 stroma. Concentrations of inhibitors were 100 nM for venetoclax, S63845, PTC596 and trametinib, 1 μM for bimiralisib, C-188-9, and A133825. Significance was calculated in a graph pad prism using grouped analysis with multiple unpaired *t*-test comparing cell viabilities of treated cells grown in the absence or presence of HS-5 stroma. Significance denoted for *p* < 0.05 (*); *p* < 0.005 (**); *p* < 0.0005 (***); no significance denoted for *p* > 0.05 (ns). A significance level of 0.05 indicates a 5% risk of concluding that a difference exists when there is no actual difference.

**Figure 4 ijms-23-12587-f004:**
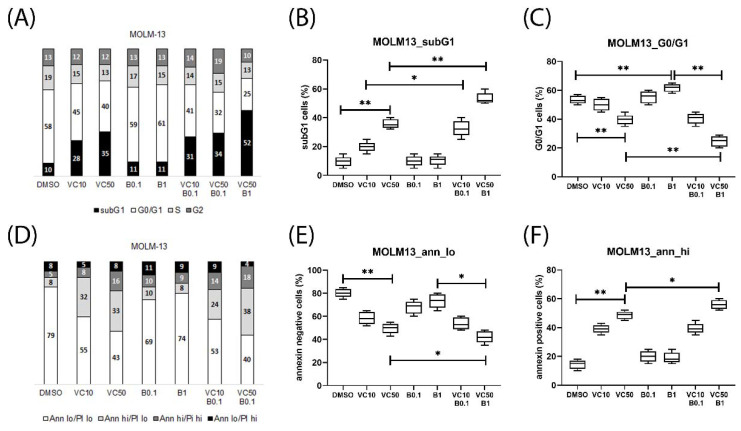
Induction of apoptosis, cell cycle arrest and cell death in MOLM_13 cells treated with venetoclax and bimiralisib. Cytometric analysis of MOLM-13 cells treated for 20 h with 10 nM or 50 nM venetoclax (VC) and 0.1 μM or 1 μM bimiralisib alone or in combination. Depending on DAPI staining intensity cells were classified as subG1, G0/G1, S phase, or G2 phase (**A**). Treatment-induced cell death (subG1 fraction) (**B**), and G1 cell cycle arrest (**C**). Depending on Annexin V and PI staining intensity, cells were classified as vital (Ann lo, PI lo), early apoptotic (Ann hi, PI lo), late apoptotic (Ann hi, PI hi) or necrotic (Ann lo, PI hi) (**D**). Treatment-induced loss of vital cells (**E**) and amount of apoptotic cells (**F**) were significanty enhanced in the combination treatment. Significance of differences in median values was calculated by the Mann–Whitney test. Significance denoted for *p* < 0.05 (*); *p* < 0.005 (**); no significance denoted for *p* > 0.05.

**Figure 5 ijms-23-12587-f005:**
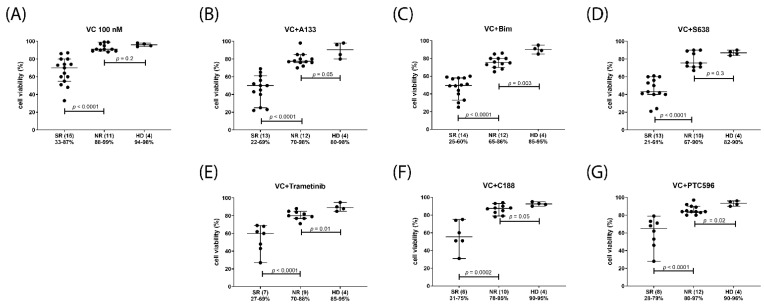
Hematological cells in vitro response to venetoclax and various combination treatments. Cell viability was determined in mononuclear cells isolated from AML patients or healthy donor (HD) peripheral blood or bone marrow after 20 h treatment. The patient samples were sorted into two equally sized groups, one with major (strong) response (SR) and one with minor (normal) response (NR). Number of samples in each group are indicated in parentheses. Cells were treated in vitro with 100 nM venetoclax (VC) only (**A**), 100 nM venetoclax in combination with 1 μM A1331825 (**B**), 1 μM bimiralisib (**C**), 100 nM S63845 (**D**), 100 nM trametinib (**E**), 1 μM C-188-9 (**F**), or 200 nM PTC596 (**G**). Significance of differences in median values was calculated by Mann–Whitney test.

**Figure 6 ijms-23-12587-f006:**
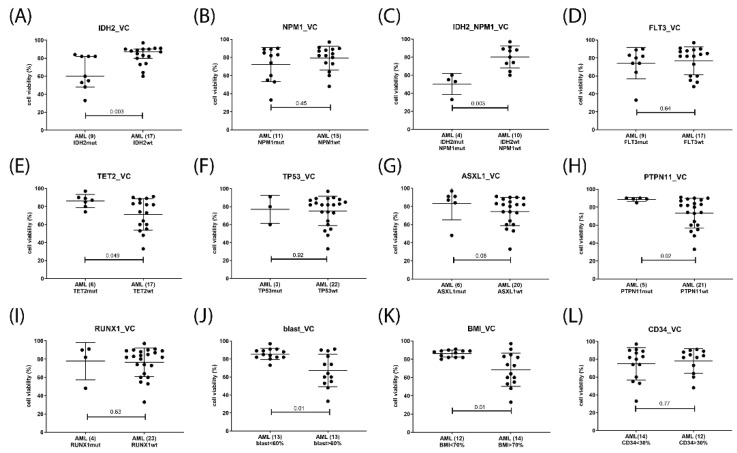
Response markers in venetoclax monotherapy. Cell viability was determined in mononuclear cells isolated from AML patients peripheral blood or bone marrow after 20 h treatment with 100 nM venetoclax. Hematological cells were grouped according to single gene mutation status of IDH2 (**A**), NPM1 (**B**), IDH2 and NPM1 (**C**) FLT3 (**D**), TET2 (**E**), TP53 (**F**), ASXL1 (**G**), PTPN11 (**H**), RUNX1 (**I**) peripheral blast cell percentage (**J**), bone marrow blast cell percentage (**K**), and CD34 positivity (**L**). Number of samples in each group are indicated in parentheses. Significance of differences in median values was calculated by Mann–Whitney test.

**Figure 7 ijms-23-12587-f007:**
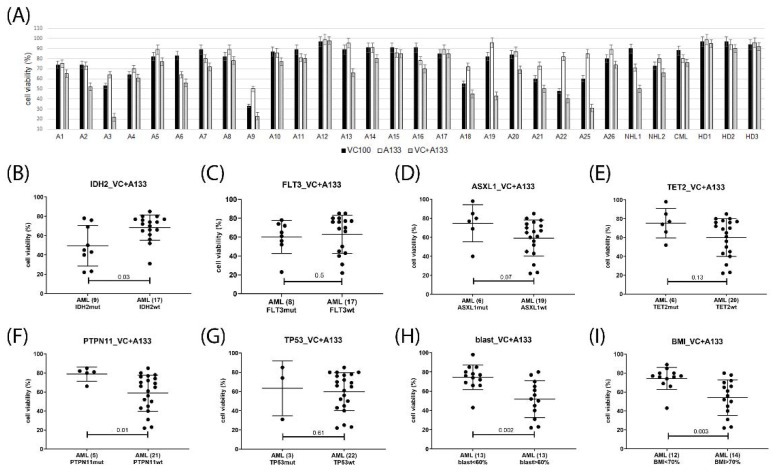
Response markers in venetoclax and A1331825 combination treatment. Cell viability was determined in mononuclear cells isolated from AML patients peripheral blood or bone marrow after 20 h treatment with 100 nM venetoclax and 1 μM A1331825 (**A**). Hematological cells were grouped according to single gene mutation status of IDH2 (**B**), FLT3 (**C**), ASXL1 (**D**), TET2 (**E**), PTPN11 (**F**), TP53 (**G**), peripheral blast cell percentage (**H**), bone marrow blast cell percentage (**I**). Number of samples in each group are indicated in parentheses. Significance of differences in median values was calculated by Mann–Whitney test.

**Figure 8 ijms-23-12587-f008:**
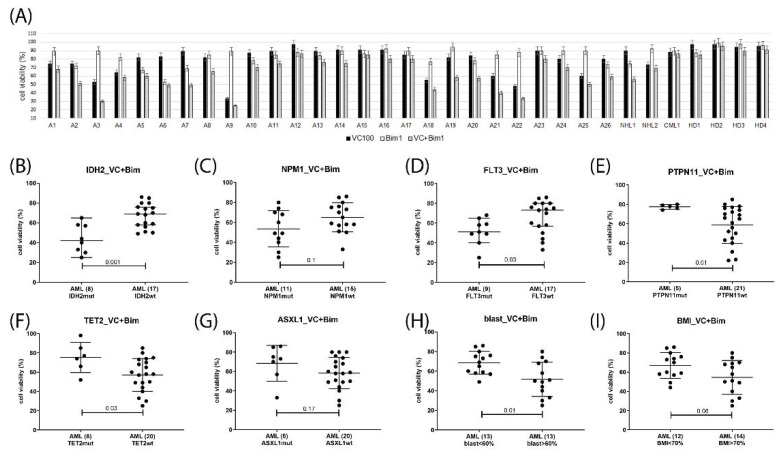
Response markers in venetoclax and bimiralisib (PQR309) combination treatment. Cell viability was determined in mononuclear cells isolated from AML patients peripheral blood or bone marrow after 20 h treatment with 100 nM venetoclax and 1 μM bimiralisib (**A**). Hematological cells were grouped according to single gene mutation status of IDH2 (**B**), NPM1 (**C**), FLT3 (**D**), PTPN11 (**E**), TET2 (**F**), ASXL1 (**G**), peripheral blast cell percentage (**H**), and bone marrow blast cell percentage (**I**). Number of samples in each group are indicated in parentheses. Significance of differences in median values was calculated by Mann–Whitney test.

**Figure 9 ijms-23-12587-f009:**
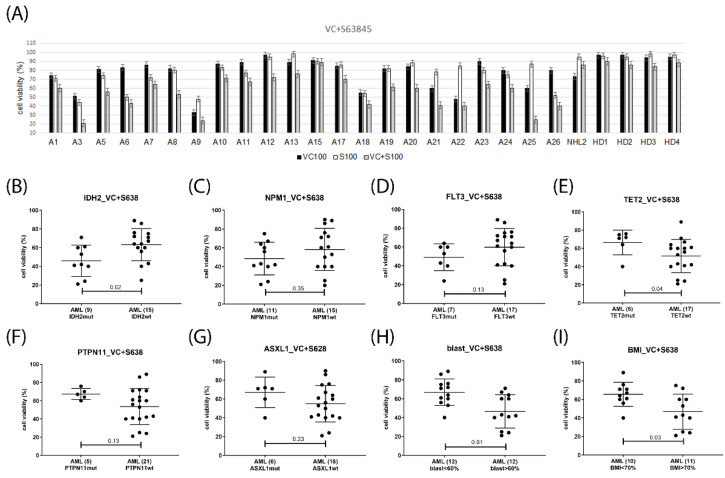
Response markers in venetoclax and S63845 combination treatment. Cell viability was determined in mononuclear cells isolated from AML patients peripheral blood or bone marrow after 20 h treatment with 100 nM venetoclax and 100 nM S63845 (**A**). Hematological cells were grouped according to single gene mutation status of IDH2 (**B**), NPM1 (**C**), FLT3 (**D**), TET2 (**E**), PTPN11 (**F**), ASXL1 (**G**), peripheral blast cell percentage (**H**), and bone marrow blast cell percentage (**I**). Number of samples in each group are indicated in parentheses. Significance of differences in median values was calculated by Mann–Whitney test.

**Table 1 ijms-23-12587-t001:** Characteristics of leukemia and lymphoma cell lines.

ID	Disease	Status	*FLT3*	*TP53*	Gene Variants	Karyotype
HL-60	AML (M2)	de novo	wt	null	NRAS Q61L CDKN2A R80X	hypotetraploid
ML-2	AML (M4)	de novo	wt	wt	KMT2A-AFDN KRAS A146T	t(6;11)
MOLM-13	AML (M5)	relapse	ITD	wt	KMT2A-MLLT3	t(9;11)
MOLM-16	AML (M0)	relapse	wt	V173M/C238S	MLL V1368L	hypotetraploid
OCI-AML3	AML (M4)	de novo	wt	wt	DNMT3A R882C NRAS Q61L NPM1 L287fs	+1, +5, +8
PL-21	AML (M3)	de novo	ITD/P336L	wt/P36fs	KRAS A146V	hypertetraploid
SKM-1	AML (M5)	refractory	wt	R248Q/R248Q	ASXL1 Y591Ter KRAS K117N	del(9q12)
OCI-Ly1	DLBCL	relapse	wt	R158H/C176G	BCL2-IgH, PTEN del	t(14;18)

AML, acute myeloid leukemia (FAB classification); DLBCL, diffuse large B-cell lymphoma; wild type (wt); internal tandem duplication (ITD).

**Table 2 ijms-23-12587-t002:** IC50 values cell lines (μM).

	Targeted Therapy						
Cell Line	Venetoclax	A1331825	PQR-309	C-188-9	PTC596	S63845	Trametinib
Target	BCL-2	BCL-XL	PI3K, mTOR	STAT3	BMI-1	MCL-1	MEK
HL-60	1	4	5	5	0.2	0.1	0.08
ML-2	0.08	2	3	4	1.5	0.5	0.12
MOLM-13	0.1	6	2	4	0.3	0.01	0.12
MOLM-16	>10	2	10	>10	1.1	10	10
OCI-AML3	0.2	4	10	8	0.5	0.2	0.1
PL-21	10	2	10	>10	0.8	1	10
SKM-1	2	8	3	8	1.2	0.5	0.12
OCI-Ly1	0.06	8	1	8	1	0.12	0.3

**Table 3 ijms-23-12587-t003:** Combination index values of AML cell lines.

	Venetoclax Combination Treatment					
Cell Line	A1331825	Bimiralisib	C188-9	PTC596	S63845	Trametinib
HL-60	0.6–0.8	0.2–0.4	0.4–0.6	0.2–0.4	0.4–0.6	0.2–0.4
ML-2	0.3–0.5	0.3–0.5	0.9–1.1	0.8–1.0	0.1–0.3	0.3–0.5
MOLM-13	0.2–0.4	0.3–0.5	0.9–1.1	0.6–0.8	0.2–0.4	0.2–0.4
MOLM-16	0.2–0.4	>1.1	0.3–0.5	0.7–0.9	0.7–0.9	0.9–1.1
OCI-AML3	0.2–0.4	0.5–0.7	0.8–1.0	0.6–0.8	0.2–0.4	0.3–0.5
PL-21	0.4–0.6	0.8–1.0	0.5–0.7	0.9–1.1	>1.1	0.9–1.1
SKM-1	<0.1	0.3–0.5	0.7–0.9	0.3–0.5	<0.1	<0.1
OCI-Ly1	0.4–0.6	0.6–0.8	0.7–0.9	0.8–1.0	0.4–0.6	nd

Combination indexes (CI) were calculated according to Chou Talalay [38]. Interpretation: CI = 0.1–0.3 strong synergy, CI = 0.3–0.7 moderate synergy, CI = 0.7–0.9 mild synergy, CI = 0.9–1.1 additive effects, CI > 1.1 antagonistic effects.

**Table 4 ijms-23-12587-t004:** Clinical characteristics of hematological samples.

ID	Disease	Mutation Profile	Cytogenetics	Source	PBC	BMI	CD34+
					%	%	%
AML1	AML-M1	FLT3-ITD (>1), NPM1	normal	PB	90	90	5
AML2	AML-M1	FLT3-ITD (0.78), U2AF1, BCOR, TET2	del(20)(q11.2q13), +8	PB	62	90	18
AML3	AML-M5a	NPM1, IDH2	normal	BM	87	90	18
AML4	AML-M4	FLT3-TKD, KMT2A-MLLT10	t(10;11)	PB	88	80	45
AML5	AML-M4	NPM1, DNMT3A, NF1	normal	BM	8	20	11
AML6	AML-M5	NPM1, FLT3-TKD (0.63), DNMT3A	normal	BM	86	95	2
AML7	AML-M5	NPM1, FLT3-ITD (0.58), DNMT3A	normal	PB	1	70	7
AML8	AML sec	FLT3-ITD, IDH2, RUNX1, DNMT3A	tetraploid, del5q	BM	45	80	56
AML9	AML-M1	NPM1, FLT3-ITD (9.45), IDH2	normal	PB	94	90	20
AML10	AML-M4	ASXL1, TET2, KRAS	normal	BM	nd	30	1
AML11	AML-M4	NPM1, PTPN11	normal	BM	65	65	22
AML12	AML-M5	ASXL1, TET2, KRAS, SH2B3, U2AF1	mono7	PB	53	80	30
AML13	AML sec	TET2, DNMT3A, PTPN11	mono7, del(12), inv(9)	PB	36	50	90
AML14	MDS-AML	CEBPA, ASXL1, EZH2, RUNX1	normal	BM	13	nd	26
AML15	MDS-AML	ASXL1, TP53, CALR	KMT2A amp (97%)	PB	20	20	52
AML16	AML-M1	normal	mono9, 11q23.3	PB	96	90	38
AML17	AML-M4/5	NPM1, DNMT3A, TET2, PTPN11	normal	PB	19	90	68
AML18	AML-M1	NPM1, IDH2, SRSF2	normal	PB	72	70	1
AML19	AML-M2	IDH2, DNMT3A	der(16)t(11;16), +14	BM	33	60	90
AML20	AML-M4	ASXL1, IDH2, DNMT3A, SRSF2	normal, +8	BM	9	50	82
AML21	AML-M2	NPM1, IDH2	normal	BM	95	90	24
AML22	AML-M0	ASXL1, IDH2, RUNX1	normal	BM	64	80	94
AML23	AML-M2	RUNX1, TET2, PTPN11, PRPF8, NF1	mono7, t(9;22)	PB	68	45	82
AML24	AML sec	FLT3-TKD, IDH1, NPM1, PTPN11, SRSF2	normal	BM	83	90	1
AML25	AML-M1	TP53	complex	PB	75	80	97
AML26	AML-M4	FLT3-TKD (0.56), TET2, SRFS2, TP53	normal, +8	PB	46	50	10
CML1	CML	BCR-ABL1	t(9;22)	PB	1	5	42
NHL1	NHL	TP53	normal	PB	nd	nd	nd
NHL2	NHL	cMyc and BCL2 rearranged (double hit)	t(8;14), t(14;18)	BM	18	89	58

AML, acute myeloid leukemia; CML, chronic myeloid leukemia; HD, healthy donor, NHL, non-Hodgkin lymphoma. FLT3 gene mutant allele ratio is indicated in parentheses. PBC, peripheral blast count; BMI, bone marrow infiltration.

## Data Availability

Data are contained within the article or Appendix A.

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
