# Peer review of "Rationale for Combining the BCL2 Inhibitor Venetoclax with the PI3K Inhibitor Bimiralisib in the Treatment of IDH2- and FLT3-Mutated Acute Myeloid Leukemia"

_ijms, 2022, doi:10.3390/ijms232012587_

Round 1

Reviewer 1 Report

Very interesting topic and results. However, following steps of analyses and selection of cell lines/treatment need to be clearly stated in the manuscript. As the outcomes are clinical, a brief description of the all tests/procedures currently recommended to be performed at diagnosis and selection of therapy should be provided.

1.       The results of the cytotoxicity assays in Subsection 2.1. should be presented in a table. In the Materials and Methods section, specify what concentrations of venetoclax and investigated inhibitors were tested at this stage.

2.       Why were only 4 cell lines investiagted in Subsection 2.3.? The description of Figure S1 is unclear and inadequate. Why did the authors decide to include this part of the results in the Supplementary file?

3.       Why in Subsection 2.4. only 2 cell lines and treatment with bimiralisib and venetoclax (alone an in combination) were investigated?

4.       In subsection 2.4, the Authors state:

"(...) The tested combination treatments induced minor reduction of cell viabilities in mononuclear cells isolated from healthy donors, and substantial reduction of cell viability in 50-70% of AML samples treated with venetoclax alone or in combination with A1331825, S63845, bimiralisib or trametinib (Fig. 4) "

There is no information regarding the BMI-1 inhibitor. Simultaneously, the result for C-188-9 is not presented in Figure 4. There are errors in the Figure 4 description.

5.       What criteria were adopted to sort the patient samples into NR and SR groups?

6.       Statistical analyses were performed with a Mann-Whitney test. What significance level was used for the interpretation of the obtained p value?

7.       The number of subjects in particular subgroups included into statistical analysis should be given in the Figures (Figure 4, 5, 6, 7 and 8).

8.       The Authors did not take into account the limitations of a presented study.

9.       What additional molecular tests do the authors recommend among AML patients? What mutations/gene regions should be assessed in order to apply the venetoclax combination treatment proposed in this manuscript?

Reviewer 2 Report

This article by Katja Seipel et al., represents a nice study of AML cells' sensitivities to various therapeutic agents when treated in the dish. The paper is well-written and does not need, in my view, significant edits or improvements. The only comment or suggestion I have is if the authors could add a short paragraph (a couple of references) about similar in vitro studies that subsequently used PDX treatment models, and then translated to clinical studies involving human subjects. I am sorry if I overlooked that information and it is already in the paper, but the study would benefit from highlighting the next steps of translational research. Also, relevance to the physiological dosage of these drugs, and their combinations, is worth mentioning. 

Reviewer 3 Report

Journal: International Journal of Molecular Sciences (ISSN 1422-0067).

Manuscript ID: ijms-1916794

Title: Rationale for combining the BCL2 inhibitor venetoclax with the PI3K inhibitor bimiralisib in the treatment of IDH2 and FLT3 mutated acute myeloid leukemia

Special Issue: Advances in Molecular Pathogenesis and Targeted Therapies for Myeloid Neoplasms

Seipel et al. present an interesting study dealing with the evaluation of different inhibitors as single agents and in combination with the BCL2 inhibitor venetoclax. Although the topic is of potential interest the results appear to be too incomplete and the manuscript contains some fundamental flaws. Generally better workup of the data is necessary that readers can fully recapitulate and profit from this study.

Major points:

2.1

a.       It is of general interest to test the inhibitors on different AML cell lines. But the figure/data is missing. It needs to be shown, how the IC50 was defined/measured.  

b.       Reference of Table 1 is missing. Abbreviations are incomplete.

c.        Figure 1: Abbreviations are missing. Description is incomplete. Colors are not well chosen. 

2.2

a.       How did the authors decide which concentration they use for this experiment? – this is not conclusive and needs to be stated in the manuscript.

b.       Unnecessary self-citation! If the authors claim to calculate “according to Chou Talalay“ they should cite Chou T.-C. Drug Combination Studies and Their Synergy Quantification Using the Chou-Talalay Method. Cancer Res. 2010;70:440–446. doi: 10.1158/0008-5472.CAN-09-1947!

c.        Figure 2: Statistics are missing. Abbreviations are missing

d.       Table 2: Are the values synergistic effects as mentioned in the text, then the table needs to be improved, because this is not clear. Or is it Table 3?

e.       In the Table 3 legend is written: “Combination indexes were calculated according to Chou Talalay [28].“ But reference 28 is: Redell, M.S.; Ruiz, M.J.; Alonzo, T.A.; Gerbing, R.B.; Tweardy, D.J. Stat3 Signaling in Acute Myeloid Leukemia: Ligand-Dependent and -Independent Activation and Induction of Apoptosis by a Novel Small-Molecule Stat3 Inhibitor. Blood 2011, 117, 5701–5709, doi:10.1182/blood-2010-04-280123.   

Why are there only 4 cell lines listed in Table 3?

2.3

a.       Here again, references are missing.

b.       it is important to show the activation (and subsequent inhibition) of relevant pathways (e.g. STAT signaling) which get activated by G-CSF. GM-CSF, M-CSF, IL-1alpha, IL-1beta, IL-1RA, IL-6, IL-8, IL-11, and LIF with Western blot.

c.        Figure S1: Statistics are missing. Abbreviations are missing.

2.4

a.       It is not clear why the authors focus in 2.4 only on bimiralisib and venetoclax treatment in MOLM-13 and SKM-1 cells. Please show the data for other drugs and cell lines.

b.       Figure 3: Statistics are missing. Abbreviations are missing. Typos in figure labelling. For the reader its easier to use cell cycle phase instead of “ann lo/Pi lo”. DAPI is not the same as Hoechst 33342 (typo in the methods: Hoechst 33,342).

c.        “apoptosis and cell death”. I think this is not correct (Necrosis and Apoptosis). Please discriminate between, late apoptosis, early apoptosis and necrosis in the figure.

d.       Please show the raw blots assay and the gating strategy for cell cycle analysis and apoptosis in the supplements.

2.5

a.       There is no 2.5.

b.       Nice data, but this part shows clearly that the manuscript is incomplete. Therefor it is hard the fully comment on this part.

c.        There is almost no description for Figure 5, Figure 6, Figure 7 and Figure 8.

d.       Here again, figure legends and headlines need to be improved!

e.       Figure labelling is too small

f.         The analysis of the mutational status of the patient samples needs a better description in the method section.

g.       In addition, all samples were analyzed by NGS sequencing of the myeloid panel genes.” …  The “S” in “NGS” stands for sequencing. How was the sequencing done for myeloid panel genes?,…which instrument,…etc?

Minor points:

-          Why were median values and the Mann-Whitney test used for every dataset? Please comment on this for every experiment.

-          Figure and result titles should contain the result itself

-          Some abbreviations are not named or named at the wrong position

-          Figure labelling, resolutions and arrangement need to be improved.

-          Method section is incomplete and given information is not consistent. Some methods are not described and important information (e.g. catalog numbers, manufactures) is missing. The readers should be able fully recapitulate all the methods.

-          Some references are missing

Round 2

Reviewer 1 Report

Thank you for your reply to the comments and the applied changes. The manuscript is appropriate to be accepted as it stands; I have no further comments.

Reviewer 3 Report

The additional data and the work-up significantly improved the manuscript, however some of my points have been ignored or not been addressed sufficiently.

- Even some minor comments (e.g. Typos (NGS sequencing), DAPI is not the same as Hoechst, discriminate between late apoptosis, early apoptosis, etc.) have been completly ignored.

- In my opinion it is important to show the activation of the signaling pathways. Without this information these data would be neglectable.

- Statistics are still incomplete and, in my opinion, incorrectly applied.